

# Prioritization of the vector factors controlling *Emiliania huxleyi* blooms in subarctic and arctic seas: A multidimensional statistical approach

Dmitry Kondrik[1], Eduard Kazakov[1,2,3], Svetlana Chepikova[1], Dmitry Pozdnyakov[1]

[1]Nansen International Environmental and Remote Sensing Centre, Saint Petersburg, 199034, Russia
[2]LLC NextGIS, Moscow, 117312, Russia
[3]Russian State Hydrological Institute, Saint Petersburg, 199034, Russia

*Correspondence to*: Dmitry Kondrik (dmitry.kondrik@niersc.spb.ru)

**Abstract.** Producing very extensive blooms in the world's oceans in both hemispheres, a coccolithophore *E. huxleyi* is capable of affecting both the marine ecology and carbon fluxes at the atmosphere-ocean interface. At the same time, it is subject to the impact of multiple co-acting environmental forcings, which determine the spatio-temporal dynamics of *E. huxleyi* blooming phenomenon.

To reveal the individual importance of each forcing factor (FF) that is known to significantly control the extent and intensity of *E. huxleyi* blooms and can be retrieved from remote sensing data, we used long-term spatial time series (1998-2016) of sea surface temperature and salinity, incident photosynthetically active radiation, and Ekman layer depth relevant to the marine environments located in the North Atlantic, Arctic and North Pacific oceans, namely the North, Norwegian, Greenland, Labrador, Barents and Bering seas.

The FFs retrieved were subjected to statistical analyses. The descriptive statistical approach has shown that *E. huxleyi* phytoplankton were highly adaptive to the environmental conditions and capable of arising and developing within wide FFs ranges, which proved to be expressly sea-specific. It was also found that there were FFs optimal ranges (also sea-specific), within which the blooms were particularly extensive.

The application of the Random Forest Classifier (RFC) approach to each target sea allowed to reliably rank the FFs considered in terms of their role in the spatio-temporal dynamics of *E. huxleyi* blooms. With the only exception of the Bering Sea, allegedly due to temporally established untypical hydrological conditions, the prediction ability of RFC modeling characterized in terms of precision, recall, and *f1*-score generally was in excess of 70%, thus indicating the adequacy of the developed models for FFs prioritization with regard to *E. huxleyi* blooms.

## 1 Introduction

A unicellular marine alga *Emiliania huxleyi* is the most abundant and productive calcifying organism in the world's oceans (McIntyre and Be, 1967). It is a veritably cosmopolitan planktonic species growing at subpolar, temperate, subtropical and



tropical latitudes (Brown and Yoder, 1994; Iglesias-Rodrigues et al., 2002; Moore et al., 2012) and in the waters with the trophic status within the eutrophic to oligotrophic range (Paasche, 2002).

The property of this photosynthesizing aquatic organism to produce not only organic carbon, but also calcite, i.e. particulate inorganic carbon (PIC), imparts to *E. huxleyi* a special importance for the global ocean carbon cycle, and, through intricate

interactions, for $CO_2$ exchange between the ocean and the atmosphere (Balch et al., 2016; Kondrik et al., 2018). It implies a definite climatic dimension in the overall environmental impact of this phenomenon. This assertion is substantiated by the typical scale of the *E. huxleyi* phenomenon: the blooms are predominantly gigantic covering marine areas in excess of many hundred thousand, sometimes up to one million, square kilometers. Occurring annually across the world's oceans (Moore et al., 2012), such blooms (Brown and Yoder, 1994) not only release in water huge amounts of PIC, in some cases reaching

nearly one million tons, but also enhance $CO_2$ partial pressure in surface waters at least within *E. huxleyi* bloom areas (Kondrik et al., 2018; Balch et al., 2016).

Despite the remarkable susceptibility of *E. huxleyi* to accommodate to very different living conditions, the blooms of this alga exhibit very significant interannual variations in extent, intensity and localization (e.g. Smyth et al., 2004; Balch et al., 2012; Iida et al., 2012; Morozov et al. 2012; Kondrik et al., 2017), as well as a tendency to proliferate poleward (Winter et al., 2013).

Importantly, the aforementioned spatio-temporal variations inherent in *E. huxleyi* blooms prove to be specific to concrete marine environments, which indicates that *E. huxleyi* growth is generally conditioned by multiple factors acting through forward and feedback mechanisms.

Numerous studies (see e.g. reviews by Tyrrell and Merico, 2004; Paasche, 2002; Thierstein and Young, 2004; Zondervan, 2007; Feng et al., 2008; De Bodt et al., 2010) show that the factors affecting *E. huxleyi* blooms can be broadly divided into

two groups, viz. vector and scalar factors. In turn, the two group factors are liable to be further subdivided to specify the concrete nature of forcings and associated mechanisms of their performance. However, such a division is rather conditional as the influence of factors on *E. huxleyi* growth from one group can be indirect through bringing the other group factor(s) into play (Shutler et al., 2013).

Water temperature, salinity, alkalinty/acidity, water column stratification, water movements (currents, eddies, fronts,

advection), availability of nutrients and trace metals, viruses, microzooplankton grazing, seeding, water surface illumination, wind and wave driven surface water mixing, large-scale atmospheric baric formations and air mass transport, teleconnections, were investigated by many workers as factors capable of affecting *E. huxleyi* blooms seasonally, and what is more important, interannually (Townsend et al., 1994; Bratbak et al., 1995; Nanninga and Tyrrell, 1996; Thierstein and Young, 2004; Moore et al., 2012 and refs. therein).

Some universal prioritization of the above factors seems hardly possible as in all evidence their individual efficiency can only be assessed in laboratory experiments while in real life it is affected by concomitant factors that are able either subdue or contrarily enhance the overall impact. Moreover, as there are several strains of *E. huxleyi* (Paasche, 2002), the results obtained in meso-microcosm experiments for a certain strain are not necessarily applicable to *E. huxleyi* blooms to generalize the effect of this phenomenon in marine environments.



This can be illustrated with water temperature as one of the parameters allegedly controlling the growth of *E. huxleyi* cells. The reports on water temperature limits, at which this alga can grow are many and frequently inconsistent, e.g. $\leq 0 - >30$ °C (Balch et al., 1991), 1–30 °C (Okada and McIntyre, 1979); 2–28°C (Westbroek et al., 1993); 3–15 °C; 7–27 °C (Watabe and Wilbur, 1966), 3–25 °C (Mjaaland, 1956).

The reported data on the optimal temperature magnitude/range for *E. huxleyi* growth are also rather mutually at odds, e.g. >12 °C (Hattori et al, 2004), 11–23 °C (Haidar and Thierstein, 2001), 14–17 °C (Balch et al., 1991), (18–24 °C (Watabe and Wilbur, 1966; McIntyre and Be, 1967), 20 °C (Mjaaland, 1956; Le Vu, 2005), 20–25 °C (Haidar and Thierstein, 2001).

In light of the above, it appears logical that many authors consider the water temperature impact exclusively in conjunction with other factors, such as water salinity (Townsend et al., 1994; Buitenhuis et al., 2008), light availability and nitrate low

concentration (Tyrrel and Taylor, 1996), nutrients (Silkin et al., 2009), light and water stratification (Nanninga and Tyrell, 1996), and many others listed earlier. It can be concluded that water temperature conditionally can be assumed as a good predictor of *E. huxleyi* bloom initiation and further development, however, in close conjunction with the co-occurring external and in-water factors.

Arguably, the same conclusion can be made with respect to all other forcing factors. It implies that analyses of such a multi-

factored phenomenon need to be performed with multi-dimensional statistical means for each target water body separately.

The present study is aimed at interpreting a 19-year time series of spatial and temporal variations of *E. huxleyi* blooms in subpolar and polar marine environments reported elsewhere (Kondrik et al., 2017, Kondrik et al., 2018). Obtained from satellite ocean color data, the time series (1998–2016) of *E. huxleyi* bloom area, inorganic carbon production and dissolved $CO_2$ partial pressure exhibit significant variations that proved to be specific for each of the target seas, and do not reveal any obvious

patterns or trends. These findings, firstly, support the above stated vision of this phenomenon as a multi-factored one, and secondly, call for employment of appropriate statistical instruments for analyses and robust interpretations of the observed time-series features. To this end a multi-dimensional statistical tool was employed, viz. Random Forest Classifier (RFC).

Below we start with a justification of choice of the above tool, and its concise description. This is further followed by substantiation of the forcing factors selected in the present study. These sections precede the description of our analysis results,

their discussion and major conclusions.

## 2 Selection of forcing factors (FFs) and sources of data employed

### 2.1 FFs selection incentives

Ideally, for prioritization of factors controlling *E. huxleyi* blooms, the entire set of the FFs, which were adverted to in Sect. 1, needs to be submitted to analyses. However, since in the present study we are dealing with six target seas (the North,

Norwegian, Labrador, Greenland, Barents and Bering seas) and the overall time period of nearly two decades (1998–2016), this task is unattainable. Indeed, in our case the data liable to statistical treatment should be co-located in space and time as



well as statistically significant for the bloom phenomenon in each year. Understandably, some limitations in this regard are inevitable.

## 2.2 Sources of data on FFs

The FFs enumerated in Sect. 1 can be conditionally partitioned into two groups: FFs derivable from (*i*) in situ/shipborne, and

(*ii*) satellite remote sensing measurements. The latter group encorporates:

*Sea surface temperature* (**SST**): satellite data at a 4 km x4 km spatial and daily resolution for the time period 1998–2014 are from the Pathfinder v5.3 product based on the Advance Very High Resolution Radiometer (AVHRR) measurements (NOAA, Korak et al., 2018; Casey et al., 2010; https://data.nodc.noaa.gov/cgi-bin/iso?id=gov.noaa.nodc:AVHRR_Pathfinder-NCEI-L3C-v5.3); SST data for 2015 and 2016 are from MODIS Aqua on the NASA website https://oceancolor.gsfc.nasa.gov/);

*Sea surface salinity* (**SSS**): satellite data at a 25 km x 25 km spatial and 4-day time resolution are from the CEC-Locean L3 Debiased v3 product (Boutin et al., 2018 a, b) owned by the Salinity Expertise Centre (CECOS), Centre Aval de Traitement des Donnees SMOS (CATDS);

*Incident photosynthetically active radiation* (**PAR**): satellite data at a 4 km x 4 km spatial and 8 day temporal resolution are from the GlobColour project product (d'Andon et al. 2009; Maritorena et al., 2010);

*Ekman layer depth* (**ELD**): reflecting the degree of impact of the above-surface winds on mixing of upper layer of the ocean, was retrieved with the methodology by Dietrich & Kale (1957) from above water satellite data on wind speed (WS) at a 0.25° x 0.25° spatial and 6-hour temporal resolution are from the CCMP (cross-Calibrated MultiPlatform) Wind Vector Analysis Product v2 (Atlas et al.,2011; Wentz et al., 2015);

*Concentration of phytoplankton chlorophyll* (**CHL**): satellite data at a 4 km x 4 km spatial and 8 day temporal resolution from

the OC CCI v3.1 product (Sathyendranath et al., 2018);

*Water surface geostrophic current speed* (**CS**): satellite data at a 0.25° x 0.25° spatial and daily resolution are from the Sea Level TAC DUACS product owned by the Copernicus Marine Environment Monitoring Service (CMEMS, http://marine.copernicus.eu/services-portfolio/access-to-products/?option=com_csw&view=details&product_id=SEALEVEL_GLO_PHY_L4_REP_OBSERVATIONS_008_047);

The World Ocean Database 2013 (WOD13, Boyer et al., 2013) was used to collect in situ data on the concentrations of nitrates, phosphates, silicates, dissolved oxygen, pH and alkalinity within *E. huxleyi* blooms in the target seas over the time period 1998–2016. But despite the fact that the number of entries for each of the parameters constituted several millions, the application of respective regional and temporal filters reduced this number drastically rendering it statistically insignificant. Similarly, in relation to the *E. huxleyi* phenomenon, all open-source in situ data on such FFs as nutrients, water column

stratification, trace metals, viruses, water temperature and salinity also proved to scarce/statistically unreliable, spatially and temporally fragmentary. However, the above in situ data were used as benchmarks for the respective remote sensing data.



Examination of the presently available archives of FFs data showed that exclusively remote sensing data either on some selected FFs or their proxies complied, at least in most cases, with the requirements specified above. SST, SSS, CS, ELD, PAR, and CHL were exploited in our analyses.

As the insuperable necessity to employ solely FFs attainable from remote sensing data challenges the validity of yielded results,
special verification studies were undertaken and presented in Sect. 4.

Raw data were re-projected to the Lambert Azimuthal Equal Area Projection with the parameters corresponding to the NSIDC EASE-GRID NORTH (EPSG: 3973) coordinate system with the spatial domain whose integral area is 100,000,000 km2 at a resolution of 4 km x 4 km. The data thus processed cover the time period of 19 years (1998–2016) at a time resolution of 8 days (i.e. 874 time periods).

## 3 Methodology

### 3.1 Statistics related issues

A systematic assessment of FFs requires a general consideration of the behavioral patterns of individual environmental parameters that are (*i*) generally typical of the target body, and (*ii*) specific at the moments of observed establishment of *E. huxleyi* blooms of significant extent and intensity. To this purpose, we employed such classical methods of descriptive statistics
as boxplots (Wiliamson et al., 1989) and scatterplots (Jarrell, 1994).

Boxplots clearly display the pattern of distribution of environmental parameters, and their density within certain value ranges, whereas scatterplots reveal the nature of the interplay between two quantitative characteristics. In particular, scatterplots permit to answer such questions as (*i*) within which value ranges of one characteristic the other characteristic was observed from its inherent value range, (*ii*) how did a positive or negative dynamics of values of one characteristic relate to the respective
dynamic of the other characteristic. Collectively, both approaches allow to reveal many basic features of the data we were dealing with.

In the first place, our main interest was focused on both the specific features of behavioral patterns during *E. huxleyi* blooming, and the differences of this behavior with regard to those in bloomless and transitional periods of this alga life cycle.

Boxplots were drawn for (*i*) the samples of observation data from the bloom area, (*ii*) the area beyond the bloom area, and (*iii*)
for the entire respective sea for the entire vegetation season. This permits to reveal some salient differences in the state of the sea at the moments of *E. huxleyi* bloom intense development as well as in conditions of the bloom absence. Boxplotting also allows to identify statistically most favorable ranges of values of different FFs conditioning the target blooms. Moreover, and what is especially important, this procedure is capable of (*a*) establishing substantial area-specific differences in the conditions of bloom formation and (*b*) determining the respective ranges of FFs.

Scatterplots connect each FF with the bloom intensity as assessed via its proxy - coccolith concentration (CC, for detailed methodology see (Kondrik et al., 2017)) and thus permit to provisionally identify the important FFs, provided the condition of their connection to CC is fulfilled.





Although descriptive statistics help investigate the observed basic specific features, however, the ultimate purpose of our study was the determination of individual importance of each FF in the *E. huxleyi* blooming process. The challenge here resides in the fact that the relationship between the blooming process and each FF is not deterministic, i.e. each individual factor in the course of blooming can either strengthen or subdue the action of other FFs (Warren and Seifert, 2011).

The solution of this problem was sought in addressing approaches based on machine learning models that assure the determination of importance of each factor involved in the learning procedure. If we succeed in developing a model that successfully predicts (in terms of quality metrics) the occurrence or absence of blooming in a certain water body, then it means that the set of FFs considered in the model as well as their rating in the resultant structure of the model prove to be correct and sufficient for the target process description. To make this approach statistically more robust, the learning procedure should be

iterative and run with different observation samples containing different FFs sets in various combinations. The results thus obtained can be assumed reliable if they are reproducible with a variety of independent samples.

Within the framework of the present study the RFC algorithm of machine learning was exploited (Liaw and Wiener, 2002). We used the available realizations in the *scikit-learn library* for Python (Pedregosa et al., 2011).

The RFC algorithm is ideologically appropriate for studying the phenomenon of *E. huxleyi* blooms as it is well suited for

searching the regularities inherent in the multidimensional space of factors that mutually exert influence on each other in a sophisticated way. The efficiency of RFC application to this class of tasks is well proven by Cutler et al. (2007). Given that the preliminary tests also confirmed the preference of RFC, it was used in our study.

### 3.2 Input data related issues

As the FFs considered can exert their influence upon *E. huxleyi* blooms not solely during the blooming, but also prior to it, we

used in our analyses also FFs values recorded two and four weeks before the blooming onset. This permitted to widen the data sets, and the statistical analyses were conducted separately for each of the above time periods. In particular, one of our incentives was to evaluate the statistical importance of CHL to the bloom formation process taken two and four weeks prior to the bloom. This incentive was based on the idea that *E. huxleyi* blooms could be connected to the diatom blooms appearing in the same aquatic regions and depleting nutrients, thus "chemically preparing" the ambient water for further blooms of

coccolithophores (Lavender et al., 2008). However, our preliminary analysis has found that the above mentioned CHL influence was very low in all target seas and time periods. For that reason, the CHL data have been removed from further statistical analyses in order to improve the modeling quality of other FFs.

In addition, some input environmental parameters have not had a complete coverage of the entire time period considered in this study. For example, SSS data related only to the period 2010–2016.

Due to such data shortcomings, two time periods were selected for statistical modeling of FFs: 1. 2010–2016 and 2. 1998–2016, for which, respectively, the data on SST, PAR, CS, ELD, SSS data; and SST, PAR, CS, ELD were available of the required quality (i.e. data size and continuity, etc.).



Given that the interannual dynamics of *E. huxleyi* blooms is sea-specific and the above time periods differ in terms of their length, respective individual statistical samples were formed, and each sample was divided in two subsamples for training and testing.

When forming such subsamples, the following three conditions were observed: (*1*) training and learning samples never

overlapped; (*2*) the testing and training subsamples included all years with at least one maximal, minimal and medium blooming intensity, and (*3*) the ratio of subsample sizes constituted about 80 % and 20 %, respectively (provided the data complied with the conditions specified above and their size was sufficient).

Samples were further subjected to RFC processing to establish most robustly the model ranking of the set of FFs that determine either the bloom occurrence or its absence. The model efficiency was assessed via the metrics of its quality: precision, recall,

and *f1*-score (Eqs. (1,2); Chinchor, 1992):

$$Precision = \frac{TP}{TP+FP} \tag{1}$$

$$Recall = \frac{TP}{TP+FN}, \tag{2}$$

where TP = true positive, FP = false positive, FN = false negative, and *f1*-score = weighted average of precision and recall.

To increase the quality metrics of the best model (i.e. the model with the best prediction capacity) 20 or more iterations

(depending on the difference between the interim results) were run, and five most successful models were chosen. As the method employed does not assure equally precise results for each iteration, ranges of importance of each FF were taken from the five iterations. At the final stage, the model exhibiting the best quality metrics was identified. It is worthwhile to mention that the quality metrics of the selected five models did not differ significantly (less than 5 %). This made the determination of the ranges of parameters importance quite reliable.

**4 Results**

**4.1 Results from descriptive statistical analysis**

The starting studies employing boxplots permitted to reveal some basic statistical distribution features of the FFs considered. Two kinds of plots were generated. One of them allowed to establish separately the statistical distribution of each factor values inherent in a specific location. That was done for the entire set of satellite observations relating exclusively to the months

typical of *E. huxleyi* bloom occurrence and development. All values were partitioned into three categories: parameter values retrieved from the pixels within (*i*), and beyond (*ii*) the bloom, and (*iii*) the entire sea surface. Figure 1 exemplifies such plots for SST in the target seas, and reveals both significant and low differences in SST value ranges for categories (*i*), (*ii*) and (*iii*), respectively. For instance, the category (*i*) SST ranges in the Labrador and Bering seas differ by about 10 °C, whereas their SST category (*iii*) ranges depart from each other only by 2–3 °C. Table 1 illustrates the box and whisker ranges determined





for the category (*i*), and is indicative of a high *E. huxleyi* adaptive capacity with regard to the environmental conditions inherent in the six locations considered in our study.

The second type of plots was intended to visualize the FFs values at the dates of peak (highest extent) blooms for 1998–2016 making use of the bloom mask pooled over the entire observation period. Such plots permit to assess the interannual dynamics

in peak blooms and compare the condition under which they developed. Figure 2 illustrates the comparative FFs influence for the years of remarkably extensive and low extent *E. huxleyi* blooms in the target seas. It is noteworthy that in some instances blooms of high extent occurred at relatively low values of some FFs (e.g. SST in the North Sea, PAR and SSS in the Norwegian Sea); however, there were cases when *enhanced* values of *all* impact factors favored extended blooms (e.g. in the Greenland and Barents seas).

The above apparent inconsistencies reside in the plurality of co-acting impact factors, the complexity of their mutual feed-forward and feed-back interactions, and leaving out other impact factors that are not retrievable from spaceborne data. The employed descriptive statistics allows investigating the diversity of the FFs considered, under which the *E. huxleyi* blooming outbreaks and develops. Nevertheless, descriptive statistics is not a proper tool to establish the priority of FFs according to the degree of their influence upon the onset and dynamics of blooming. This task is within the competence of machine learning

approach, which application results are discussed in the next section.

In addition to boxplots, the scatterplots of bloom intensity (assessed via the concentration of coccoliths) were developed for all target seas and the impact factors considered in order to gain a better insight into the conditions of blooming formation. Having contours with one or two dome-like features, the scatterplots are explicitly indicative (Fig. 3 for SST) that apart from the co-occurrence of specific values of the FFs that are necessary for the blooming onset, there is a certain sea-specific range

of optimum factors values assuring enhanced intensity of coccolith production.

### 4.2 Results from modelling

Application of RFC to the satellite data relating to two time periods 1998–2016 and 2010–2016 resulted in a set of models, which were further subjected to a procedure of selection as described in Sect. 3. The selection permitted to identify five the most successful of them in assessing the importance of each FF for *E. huxleyi* blooming formation. The importance ranges of those five models are shown in Table 2 for all target seas. As seen, in the overwhelming number of cases the quality metrics

proved to be in excess of 70 %. The FF or a group of them determining the occurrence or absence of blooming proved to be sea specific and generally remained the same for the two time periods, although there several exceptions. Thus, in the case of the Norwegian Sea in the shorter time period (2010–2016) ELD proved to be the highest prioritized FF (~50 %), whereas for the extended time period (1998–2016) the importance of ELD drops down to 9–12 % ceding the "palm" to SST and PAR. In

the case of the Bering Sea, during the shorter time SSS acquired the highest priority (40 %), but in the longer time period (possibly because of lack of respective SSS data) the priority proved to be shared between a number of FFs with lower values of quality metrics.





A similar analysis was performed for a wider set of satellite observations encompassing those covering two and four weeks prior to the bloom onset. The respective results are illustrated in Table 3. It is seen that generally the FFs prioritization did not change. However, the quality metrics exhibits a wide spread of values, which implies insufficient reliability of the prioritization attained. To at least partially overcome this problem, the importance ranges in Table 3 are supplemented with the FF
importance values averaged over the five models.

Table 3 is indicative that generally the inclusion of bloom preceding data does not change the FFs prioritization order, however, there are exceptions. Thus, in the case of the Bering Sea, the quality metrics values determined with the models developed for the longer period and a two to four week time extension are much better (by about 5–12 %) reaching on average the prediction ability up to 70 %, with CS and SST retaining their leading position.

In the case of the Norwegian Sea, the predictability of the models trained with the data encompassing also the bloom preceding retrievals of FFs increases for both time periods on average by 7–8 % in comparison with the models trained without the above time extension. Importantly, for the two options the metric quality values remained very close (Table 3). SST and PAR stand out as the most significant FFs especially in the case of a four week time extension. Along with SST and PAR, the model also ranks ELD as a third runner, but in the case of time extension its influence drops down by 2.5 times.

In the case of the Barents Sea, the results of modeling are on average better by 2–3 % for the option of a two-four week time extension, which is accompanied by some reshuffling of FFs ranking, viz. the role of SSS increases, whereas the importance of PAR becomes less. Interestingly, in the longer period the CS importance in the Bering Sea is high. These issues are further dwelled upon in Sect. 5.

As the model of machine learning was chosen by us for FFs prioritization, it was important to assess the adequacy of ranking
results via checking up the model predictability per se. Such standard metrics of quality as precision, recall and *f1*-score reflect mostly the model statistical quality. Nevertheless, the phenomenon we are dealing with is in the first place geographical. Thus, it is appropriate to assess the geographical quality of the predicted bloom spatial distribution. Such maps are illustrated in Fig.4 for the Barents, Greenland and Labrador seas. All maps reveal the same specific features (also detectable in Tables 2 and 3): (*i*) the percentage of predicted blooms is very high, and (*ii*) the marine regions, for which the actual bloom had not been
predicted, are practically absent. This result evidences that our models correctly captured the natural mechanisms of the bloom onset and further development.

Also the maps in Fig.4 reveal the other feature of model predictions: the modeled bloom areas are invariably larger, sometimes significantly, than the real ones. The excessively large areas predicted by our models are located around the actual "core" of the blooming area and cover statistically homogeneous FF(s) field(s). Importantly, inaccurately predicted blooming areas are
absent in the marine parts located well away from the bloom's "core" i.e. from the area of the highest CC density.



## 5 Discussion and concluding remarks

The conducted statistical analysis revealed numerous specific features of FFs at the subpolar and polar latitudes. The descriptive statistical approach has shown that *E. huxleyi* phytoplankton were highly adaptive to the environmental conditions and capable of arising and developing within a wide range of FFs (Table 1). Although below we exemplify this thesis mostly

with SST, but our considerations are generally relevant to other FFs as well.

Despite a wide range of SST values admissible for the *E. huxleyi* bloom onset and development (up to ~20 °C), in some target seas we found significant differences in SST ranges (up to ~10 °C ) between those specifically inherent to the blooming zones, and the SST ranges (*a*) typical of the target aquatic environment during the vegetation period, and (*b*) beyond the blooming area, although in the latter case the differences did not exceed ~2–3 °C.

Also, for a small number of target seas, the dependence of the bloom area on SST could be inverse, i.e. not favoring *E. huxleyi* reproduction and spreading. For instance, in the North Sea (see Figs 2 and S2) the areas of most intense bloom occurred when the SST range was ~12–14 °C, whereas for much smaller bloom areas SST turned out to be 16 °C and even higher. At the same time, SST around 16 °C proved to be optimal for the development of the highest bloom areas in the Labrador Sea. This implies that for each individual target sea there is its specific SST optimal zones. The same was observed for the Bering Sea

(Fig. S3): the highest bloom areas occurred at SST lower by about 0.5 °C relative to the years of smaller blooms. Very similar situation was observed for SSS: although in the majority of target seas the blooms arose at ~34–35 psu, in the Labrador and Bering seas (Fig. S1) SSS was well below the background/off-bloom value.

CC (coccolith concentration) is a very appropriate proxy parameter for assessing both the quantity of inorganic carbon produced within the *E. huxleyi* bloom, and the surface water enrichment in dissolved $CO_2$. As pointed out in Sect. 1, these two

aspects of *E. huxleyi* impact on the environment are highly consequential for marine biogeochemical processes/ecology and the carbon fluxes through atmosphere-ocean interface. We found that the plots reflecting the dynamics of CC as a function of FFs had a dome-like form (Fig. 3), thus indicating that none of the FFs considered had an overwhelming individual influence on the highest bloom area development, but a collective FFs action actually determined the onset and development of peak blooms. This necessitates the application of a multidimensional analyzer such as RFC.

RFC permitted to assess not only the degree of influence of each individual FF, but to rank them for each of the target sea and reveal the proportions of respective importance. Moreover, our modeling provided pastcastings of bloom occurrence in the target seas: the retrospectively predicted bloom "cores" were in good compliance with the actually observed blooms within the test period.

The results obtained proved a veritably good prediction ability RFC modeling: practically in all cases, with the only exception

of the Bering Sea, the prediction ability expressed in terms precision, recall and *f1*-score was in excess of 70 %. In the Bering Sea, for the longer period (1998–2016) SSS was not available whereas obviously it was an FF of high influence: up to 40 % and 54 % with and without a two to four week time extension (Tables 2 and 3, respectively).



In capsule, for all target seas the quality metrics were found generally higher when extended time intervals preceding the onset of blooming were included. Only in four out of twelve cases the quality metrics were under 80 % falling in the range 70–79 % even when extended time periods were included. At that, the spread of quality metrics values showed some enhancement, although the FFs ranking order remained unaltered. The four exceptions relate to the Bering Sea (discussed above), and the

North and Norwegian seas. It can be conjected that in the last two cases the reason resides in the complexity of hydrological conditions inherent in these seas due to both their location in the way of the Gulf Stream and its ramifications and the influence of Arctic waters. Closing up this issue it appears important to underline that the quality metrics were fairly high (even up to 94 % for the Greenland Sea), none of the cases considered showed 100 %. We explain it by a limited number of FFs encompassed by our statistical analyses. Thus, the concentration of nutrients was not considered because of respective data

paucity (see Sect. 2.2). Although the inclusion of the extended time periods prior to the blooming onset was intended to account for the effects of repartition between nitrogen and phosphorous due to the preceding bloom of diatoms, certainly it was not enough to reflect the status of nutrients and their cumulative impact on *E. huxleyi* development.

Enhanced importance of CS (68 %) in the longer period is probably driven, inter alia, by specific conditions established in the Bering Sea during 1998–2001 when anomalous outbreak of *E. huxleyi* blooms took place (Kondrik et al., 2017). Namely during

the above time period, the water surface circulation pattern underwent through very significant changes (Panteleev et al., 2012), which in turn could impart additional importance to CS. With this in mind, a dropdown of the model quality metrics in the longer period can be explained by the high importance of SSS with respect to the short period. It can be hypothesized that had there been SSS data over the entire study period, SSS and CS would have been the most important ones, i.e. with the highest quality metrics values across the longer period.

It is worth mentioning that when comparing both time periods, in the longer period the quality metrics were nearly invariably higher or remained unchanged (in the case of the Barents Sea), with the only exception of the Bering and Greenland seas. The respective explanations for the Bering Sea were given above. Regarding the Greenland Sea, the explanation possibly resides in the temporal dynamics of *E. huxleyi* blooms (Kondrik et al., 2017). Indeed, in the case of Greenland Sea, the short period (2010–2016) was marked by the most extensive blooms in this sea, whereas in the previous years *E. huxleyi* blooms were

significantly less extensive and, in some years, (e.g. in 1998 and 2007) were even absent. This could lead to high noise pollution. Thus, SST proved to be most highly ranked for both time periods. It is worthwhile to add that the quality metrics also rise when the extended time data are included into analysis.

Regarding the issue of FFs prioritization, it is important to emphasize that as in the prevailing cases   the prioritization found for the longer and shorter time periods (with inclusion and none-inclusion of extended time periods) remained unaltered, the

accuracy of the selected models is good enough especially in light of complexity of the co-acting FFs. The exceptions such as the Norwegian and Bering seas are thought to be due to insufficiency/lack of input data (as discussed in Sect. 2.2) and the formation of untypical hydrological conditions.

The assessment of the geographical quality of our models via cartographic visualizations of the past actual and pastcast bloom spatial distributions in the target seas has further confirmed their appropriateness: the marine regions, for which the actual

bloom had not been predicted, are practically absent. Although the modeled bloom areas were invariably larger, sometimes significantly than the real ones, they were located around the actual "core" of the blooming area and covered statistically homogeneous FF(s) field(s). This strongly suggests that the above inconsistencies supposedly reside in the methodology of *E. huxleyi* bloom masking in satellite remote sensing data (Kondrik et al., 2017): the masked bloom has a high degree of

conformity with the contour of CC equal to 90 $10^9$ units m$^{-3}$, i.e. with the area of veritably intense blooming (named above a "core') . It implies that the actual blooming area also incorporating the core surrounding parts of the sea surface with lesser CCs is in reality larger/much larger and thus corresponds better to the modeling results. The influence of unconsidered FFs should not be either overlooked.

Although the appropriateness of utilization of the developed models for long-term projections requires further investigations,

the applicability of our models for FFs prioritization with regard to the phenomenon of *E. huxleyi* blooms seems to be quite justified.

**Author contribution**

DP, DK, and EK are responsible for the theoretical background and methodology development. DK, EK, and SCh developed the data processing algorithms, model code, and performed the simulations. All authors equally contributed to the writing of

the manuscript and data quality control.

**Competing interests**

The authors declare that they have no conflict of interest.

**Acknowledgements**

We express our gratitude for the financial support of this study provided by the Russian Science Foundation (RSF) under the

project 17-17-01117. The first author acknowledges the Nansen Scientific Society for funding his research under the Nansen Fellowship Program.

We also acknowledge with gratitude that the employed data on SST, SSS, PAR and wind were provided, respectively, by GHRSST and the NOAA National Centers for Environmental Information, the L3_DEBIAS_LOCEAN_v3 Sea Surface Salinity maps that have been produced by LOCEAN/IPSL (UMR CNRS/UPMC/IRD/MNHN) laboratory and ACRI-st

company that participate to the Ocean Salinity Expertise Center (CECOS) of Centre Aval de Traitement des Donnees SMOS (CATDS). This product is distributed by the Ocean Salinity Expertise Center (CECOS) of the CNES-IFREMER Centre Aval de Traitement des Donnees SMOS (CATDS), at IFREMER, Plouzane (France), GlobColour, ACRI-ST, France, and Remote Sensing Systems, CCMP.



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




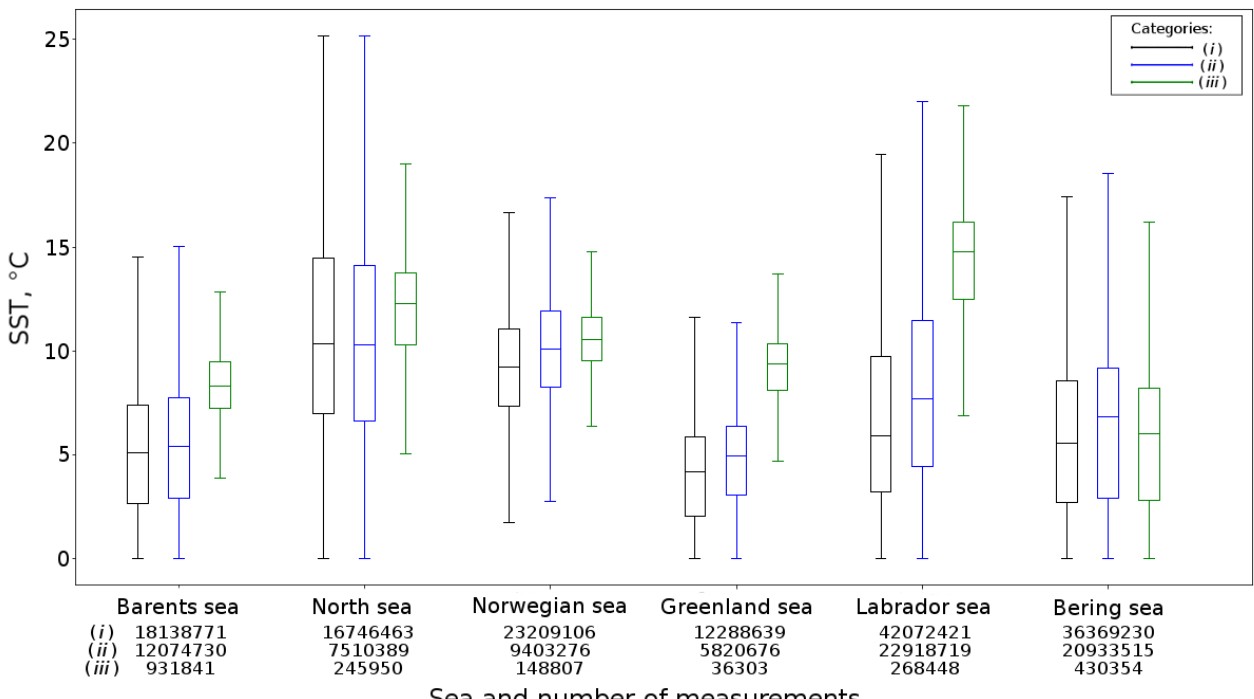

**Figure 1: Statistical distributions of SST values inherent in the target seas for all observation years. Green, blue and black colour designates categories (*i*)-(*iii*), respectively (details are in the text) The number of observations relevant to each category is specified below the horizontal axe.**



**Table 1: Ranges of FF values related to the category (*i*) of boxplots in all target seas for the period of study. Designations: * Median values, ** Box ranges, *** Whiskers ranges.**

| Sea | Parameter | | | | |
|---|---|---|---|---|---|
| | SST, °C | Salinity, psu | PAR, Ein m$^{-2}$ d$^{-1}$ | Ekman layer depth, m | Current speed, m s$^{-1}$ |
| North | 12.3* | 34.6 | 40.7 | 55.2 | 0.04 |
| | (**10.3-13.8**\*\*; 5.0-19.0\*\*\*) | (**34.1-35.0**; 32.8-36.3) | (**34.1-45.4**; 17.3-59.6) | (**47.5-65.3**; 20.8-92.0) | (**0.03-0.07**; 0.00-0.13) |
| Norwegian | 10.6 | 35.0 | 32.6 | 44.6 | 0.05 |
| | (**9.5-11.6**; 6.4-14.8) | (**34.6-35.4**; 33.6-36.4) | (**26.6-39.5**; 7.3-58.4) | (**39.3-53.7**; 17.7-75.4) | (**0.03-0.08**; 0.00-0.16) |
| Greenland | 9.4 | 34.7 | 28.6 | 35.6 | 0.04 |
| | (**8.1-10.4**; 4.7-13.7) | (**32.4-35.3**; 28.6-36.5) | (**23.5-33.5**; 8.7-48.5) | (**23.5-46.8**; 6.2-78.1) | (**0.02-0.06**; 0.00-0.13) |
| Barents | 8.3 | 34.8 | 19.8 | 49.6 | 0.03 |
| | (**7.2-9.5**; 3.9-12.8) | (**34.4-35.2**; 33.1-36.4) | (**15.8-25.1**; 1.7-39.2) | (**41.9-57.8**; 18.1-81.6) | (**0.02-0.05**; 0.00-0.09) |
| Labrador | 14.8 | 32.3 | 34.5 | 56.9 | 0.05 |
| | (**12.5-16.2**; 6.9-21.8) | (**31.8-32.7**; 30.5-34.0) | (**27.3-39.7**; 8.8-58.1) | (**48.3-68.0**; 18.9-97.5) | (**0.03-0.09**; 0.00-0.17) |
| Bering | 6.0 | 32.0 | 19.5 | 65.5 | 0.05 |
| | (**2.8-8.2**; 0.0-16.2) | (**31.7-32.4**; 30.5-33.6) | (**13.0-29.2**; 0.0-53.5) | (**52.9-79.0**; 14.3-118.3) | (**0.03-0.07**; 0.00-0.14) |





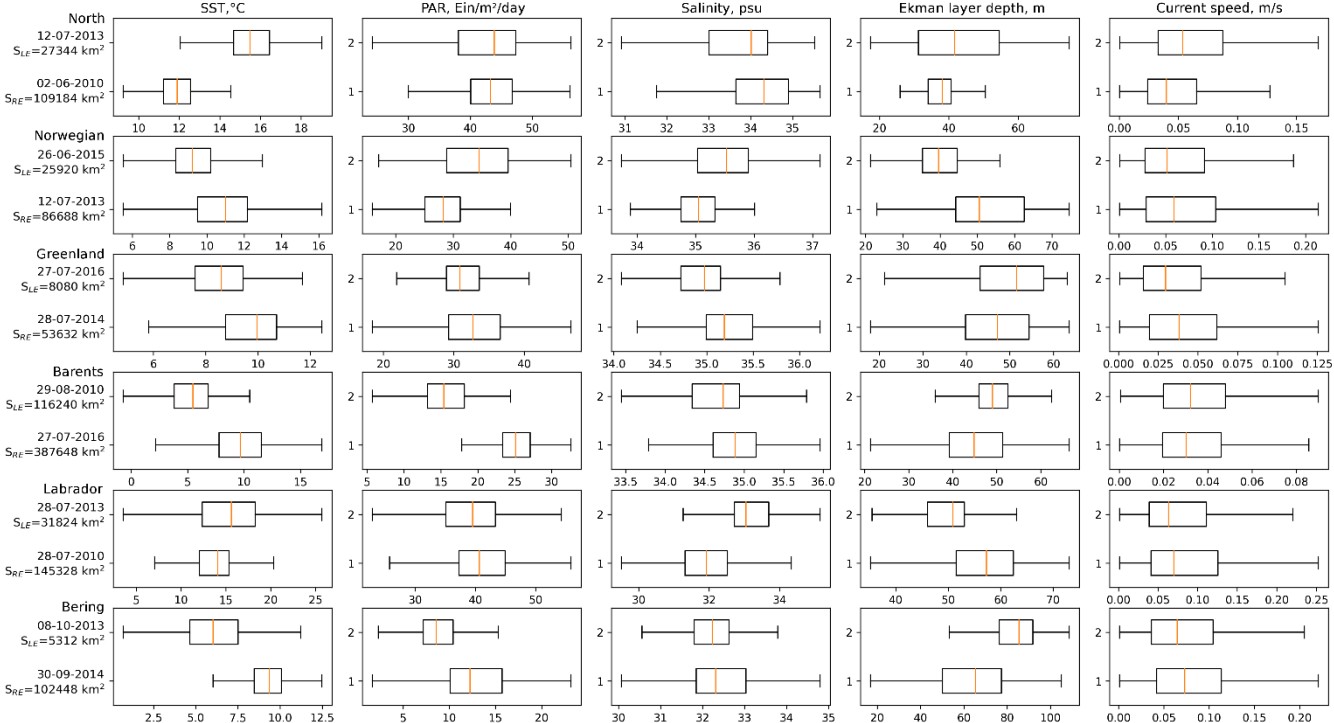

**Figure 2: A comparison of boxplots relevant to the FFs value ranges for the dates of *E. huxleyi* remarkable and low extent blooms in the target seas. Respective dates and bloom surfaces (marked as RE and LE) are specified.**





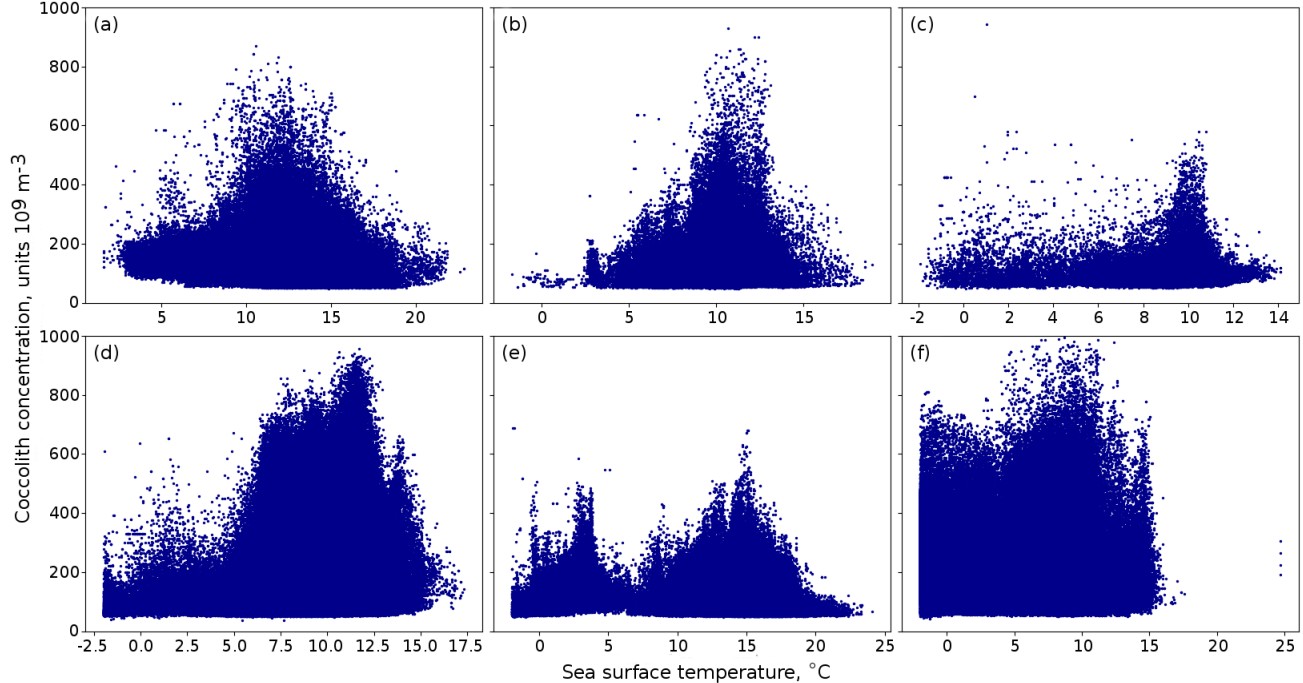

**Figure 3: CC scatterplots in dependence on SST in the North (*a*), Norwegian (*b*), Greenland (*c*), Barents (*d*), Labrador (*e*), and Bering (*f*) seas.**




**Table 2: Results of RFC implementation for the five selected FFs in all target seas.**

| Sea / Period | Barents Sea | | Greenland Sea | | North Sea | |
|---|---|---|---|---|---|---|
| | **2010-2016** | **1998-2016** | **2010-2016** | **1998-2016** | **2010-2016** | **1998-2016** |
| **Parameter** | Importance ranges, % | | | | | |
| SST | 65-66 | 78-79 | 100 | 100 | 70 | 76 |
| SSS | 4 | - | 0 | - | 8 | - |
| PAR | 27 | 17-19 | 0 | 0 | 7 | 2 |
| ELD | 3 | 1 | 0 | 0 | 5 | 2 |
| CS | 0 | 2 | 0 | 0 | 11 | 20 |
| Precision | **0.81** (0.86; 0.76) | **0.82** (0.91; 0.73) | **0.94** (0.98; 0.89) | **0.84** (0.82; 0.85) | **0.70** (0.73; 0.66) | **0.79** (0.88; 0.70) |
| Recall | **0.80** (0.72; 0.88) | **0.80** (0.66; 0.93) | **0.93** (0.88; 0.98) | **0.84** (0.86; 0.82) | **0.69** (0.61; 0.78) | **0.76** (0.60; 0.92) |
| f1-score | **0.80** (0.78; 0.82) | **0.79** (0.76; 0.82) | **0.93** (0.93; 0.94) | **0.84** (0.84; 0.83) | **0.69** (0.66; 0.72) | **0.75** (0.71; 0.79) |
| Subsample size (train / test) | 584262/ 128992 | 1295086/ 358014 | 18866/ 8890 | 40784/ 10442 | 120048/ 34274 | 345636/ 85800 |
| **Sea** | **Bering Sea** | | **Labrador Sea** | | **Norwegian Sea** | |
| **Parameter** | Importance ranges, % | | | | | |
| SST | 13-19 | 32 | 81-100 | 91 | 32-34 | 60-64 |
| SSS | 35-40 | - | 0-13 | - | 4-5 | - |
| PAR | 25-27 | 0 | 0-2 | 2 | 2-3 | 18-19 |
| ELD | 12-13 | 0 | 0-2 | 0-1 | 48-51 | 9-12 |
| CS | 8 | 68 | 0-2 | 6 | 9-10 | 1-10 |
| Precision | **0.88** (0.98; 0.79) | **0.65** (0.73; 0.57) | **0.82** (0.88; 0.76) | **0.91** (0.97; 0.85) | **0.72** (0.75; 0.68) | **0.72** (0.73; 0.72) |
| Recall | **0.86** (0.73; 0.99) | **0.61** (0.34; 0.87) | **0.81** (0.72; 0.90) | **0.90** (0.82; 0.98) | **0.71** (0.62; 0.80) | **0.72** (0.71; 0.74) |
| f1-score | **0.86** (0.84; 0.87) | **0.58** (0.46; 0.69) | **0.81** (0.79; 0.83) | **0.90** (0.89; 0.91) | **0.71** (0.68; 0.73) | **0.72** (0.72; 0.73) |
| Subsample size (train / test) | 75482/ 16062 | 365226/ 86076 | 83814/ 21578 | 221176/ 55442 | 67850/ 12844 | 195760/ 76144 |



**Table 3: Results of RFC implementation for the five selected FFs with extended periods of satellite observations (2 and 4 weeks prior to the bloom onset) in all target seas.**

| Sea / Period | Barents Sea | | | | Greenland Sea | | | | North Sea | | | |
|---|---|---|---|---|---|---|---|---|---|---|---|---|
| | 2010-2016 | | 1998-2016 | | 2010-2016 | | 1998-2016 | | 2010-2016 | | 1998-2016 | |
| **Parameter** | Importance ranges, % | | | | | | | | | | | |
| | range | mean | range | mean | range | mean | range | mean | range | mean | range | mean |
| SST | 1-36 | 23 | 17-42 | 28 | 32-75 | 46 | 29-48 | 42 | 17-33 | 24 | 20-30 | 23 |
| SST 2 weeks ago | 10-45 | 31 | 7-27 | 17 | 14-41 | 29 | 23-35 | 28 | 11-20 | 16 | 5-13 | 10 |
| SST 4 weeks ago | 3-45 | 25 | 40-52 | 44 | 1-15 | 9 | 7-22 | 13 | 22-51 | 40 | 33-47 | 39 |
| SSS | 0-2 | 1 | - | - | 0 | 0 | - | - | 0-2 | 0 | - | - |
| SSS 2 weeks ago | 0-2 | 1 | - | - | 0-5 | 1 | - | - | 0-5 | 1 | - | - |
| SSS 4 weeks ago | 5-21 | 12 | - | - | 1-13 | 7 | - | - | 0-7 | 3 | - | - |
| PAR | 1-6 | 4 | 3-9 | 5 | 0-1 | 0 | 1-2 | 1 | 0-6 | 2 | 0-2 | 1 |
| PAR 2 weeks ago | 0-1 | 1 | 0-2 | 1 | 0-9 | 4 | 3-9 | 6 | 0-8 | 4 | 3-7 | 5 |
| PAR 4 weeks ago | 0-2 | 1 | 1-2 | 1 | 1-7 | 4 | 4-9 | 6 | 0-2 | 0 | 3-7 | 5 |
| ELD | 0-1 | 0 | 0 | 0 | 0 | 0 | 0-1 | 0 | 0 | 0 | 0-1 | 0 |
| ELD 2 weeks ago | 0 | 0 | 0 | 0 | 0 | 0 | 1 | 1 | 0 | 0 | 0-1 | 1 |
| ELD 4 weeks ago | 0-1 | 0 | 0-1 | 0 | 0 | 0 | 0-1 | 1 | 0-3 | 1 | 1 | 1 |
| CS | 0 | 0 | 0-1 | 0 | 0 | 0 | 0-1 | 0 | 5-13 | 7 | 5-9 | 7 |
| CS 2 weeks ago | 0 | 0 | 1-2 | 1 | 0-1 | 0 | 1 | 1 | 0-4 | 1 | 3-5 | 4 |
| CS 4 weeks ago | 0 | 0 | 1 | 1 | 0-1 | 0 | 1-2 | 2 | 0 | 0 | 3-5 | 4 |
| Precision | **0.85** (0.92; 0.77) | | **0.85** (0.94; 0.75) | | **0.94** (0.98; 0.91) | | **0.88** (0.90; 0.86) | | **0.74** (0.78; 0.70) | | **0.79** (0.86; 0.72) | |
| Recall | **0.83** (0.72; 0.94) | | **0.82** (0.68; 0.96) | | **0.94** (0.90; 0.98) | | **0.88** (0.85; 0.90) | | **0.73** (0.64; 0.82) | | **0.77** (0.65; 0.89) | |
| f1-score | **0.83** (0.81; 0.85) | | **0.82** (0.79; 0.84) | | **0.94** (0.94; 0.94) | | **0.88** (0.87; 0.88) | | **0.73** (0.70; 0.75) | | **0.77** (0.74; 0.80) | |
| Subsample size (train / test) | 584262/ 128992 | | 1295086/ 358014 | | 18866/ 8890 | | 40784/ 10442 | | 120048/ 34274 | | 345636/ 85800 | |
| **Sea** | Bering Sea | | | | Labrador Sea | | | | Norwegian Sea | | | |
| **Parameter** | Importance ranges, % | | | | | | | | | | | |
| SST | 0-3 | 2 | 3-8 | 6 | 39-64 | 53 | 40-53 | 46 | 9-14 | 13 | 13-25 | 19 |
| SST 2 weeks ago | 5-21 | 15 | 3-15 | 9 | 4-15 | 10 | 22-33 | 28 | 3-8 | 5 | 5-13 | 9 |
| SST 4 weeks ago | 8-18 | 12 | 12-19 | 17 | 0-12 | 4 | 3-13 | 8 | 6-49 | 27 | 19-27 | 24 |
| SSS | 9-23 | 15 | - | - | 8-24 | 16 | - | - | 0-1 | 0 | - | - |
| SSS 2 weeks ago | 12-26 | 20 | - | - | 0-19 | 8 | - | - | 0-1 | 0 | - | - |
| SSS 4 weeks ago | 10-45 | 19 | - | - | 0-9 | 5 | - | - | 0-8 | 2 | - | - |
| PAR | 3-12 | 7 | 3-6 | 5 | 0 | 0 | 0-2 | 1 | 0-1 | 0 | 1-2 | 1 |
| PAR 2 weeks ago | 0-3 | 1 | 3-6 | 4 | 0-4 | 1 | 1-4 | 2 | 4-17 | 9 | 7-14 | 11 |
| PAR 4 weeks ago | 0-1 | 0 | 0-2 | 1 | 0-3 | 1 | 2-8 | 4 | 17-32 | 22 | 23-33 | 27 |
| ELD | 0-6 | 3 | 1-4 | 3 | 0 | 0 | 0-1 | 1 | 8-21 | 15 | 2-4 | 3 |
| ELD 2 weeks ago | 0-4 | 1 | 3-4 | 4 | 0 | 0 | 0-1 | 1 | 0-11 | 4 | 1-3 | 2 |
| ELD 4 weeks ago | 0-5 | 2 | 1-3 | 2 | 0 | 0 | 1-2 | 1 | 0-7 | 2 | 2-3 | 3 |
| CS | 0-1 | 0 | 11-19 | 15 | 0-1 | 0 | 0-1 | 1 | 0-1 | 0 | 0-1 | 1 |
| CS 2 weeks ago | 0-3 | 1 | 11-26 | 17 | 0-1 | 0 | 3-6 | 4 | 0-2 | 1 | 0-1 | 0 |
| CS 4 weeks ago | 0-7 | 2 | 10-23 | 17 | 0-2 | 1 | 2-3 | 3 | 0-1 | 0 | 0-1 | 0 |
| Precision | **0.85** (0.88; 0.82) | | **0.70** (0.71; 0.70) | | **0.83** (0.86; 0.79) | | **0.92** (0.98; 0.85) | | **0.79** (0.82; 0.76) | | **0.80** (0.84; 0.76) | |
| Recall | **0.85** (0.81; 0.89) | | **0.70** (0.69; 0.71) | | **0.82** (0.77; 0.87) | | **0.91** (0.83; 0.98) | | **0.79** (0.73; 0.84) | | **0.80** (0.73; 0.86) | |
| f1-score | **0.85** (0.84; 0.86) | | **0.70** (0.70; 0.71) | | **0.82** (0.81; 0.83) | | **0.90** (0.90; 0.91) | | **0.79** (0.77; 0.80) | | **0.79** (0.78; 0.81) | |
| Subsample size (train / test) | 75482/ 16062 | | 365226/ 86076 | | 83814/ 21578 | | 221176/ 55442 | | 67850/ 12844 | | 195760/ 76144 | |





**Figure 4: Cartographic visualizations of the predicted models quality for the Barents (28.07.2001), Greenland (28.07.2010) and Labrador (14.09.2002) seas.**