# Peer review of "Figure S1: Statistical distributions of (a) SSS, (b) PAR, (c) ELD and (d) CS values inherent in the target seas for all observation years. Green, blue and black colour designates categories (i)-(iii), respectively (details are in the main text) The number of observations relevant to each category"

_Biogeosciences, 2019_

## Referee Comment (RC1) · Griet NEUKERMANS (Referee) · 13 May 2019

General comments

This paper is a useful contribution to help unravel the environmental forcing of E. huxleyi blooms in the northern hemisphere high-latitude oceans and seas. The authors have used a random forest approach applied to remote sensing data to examine the relative roles of temperature, salinity, Light availability (PAR), Ekman Layer Depth, and surface current speeds as forcing factors (FFs) of Ehux blooms in six ocean

basins/seas. The results indicate that the importance of FFs is basin-specific and overall good performance of the Random Forest Model. I would recommend publication of this work in BG, but have some specific comments that should be addressed to help improve the paper.

Specific comments

Why did you use Ekman Layer Depth as an FF instead of the Mixed-Layer-Depth (MLC)? As far as I know, Ekman Layer Depth has never been shown to be a significant factor in determining Ehux blooms, whereas MLD has. MLD products are also available at a global scale on a weekly basis, so can be easily incorporated in your statistical analyses.

Further thoughts on the list of FFs considered in the analyses: it seems to me that rather than the absolute PAR, the average PAR to which the Ehux cells are exposed in the Mixed-Layer is important to consider. This is generally referred to as the "mean PAR in the Mixed Layer: PAR_ML; See for example Eq. 3 in Lacour et al. 2015 (Lacour, L., H. Claustre, L. Prieur, and F. D'Ortenzio (2015), Phytoplankton biomass cycles in the North Atlantic subpolar gyre: A similar mechanism for two different blooms in the Labrador Sea, Geophys. Res. Lett., 42, doi:10.1002/ 2015GL064540.). Also, PAR_ML is easy to obtain on a global scale and on a weekly basis.

E. huxleyi has many morphotypes (types A, B, B/C, C, T, O, R etc.) adapted to specific environmental conditions. E.g. morphotype A is considered "warm water" , while Type C is considered "cold water". It seems to me that this is an important aspect to mention when unravelling the FFs of E. huxleyi blooms and growth, and might explain why the importance of your FFs are regionally specific.

It seems to me that the lack of any ocean carbonate chemistry parameter as a potential forcing factor in your statistical analyses is a serious limitation and this should be clearly stated in the various sections of your manuscript, foremost the abstract. Why don't you just add the Takahashi pCO2 data or similar ocean carbonate chemistry datasets (other

than the World Ocean Atlas, which you find to be insufficient)?

You have found (P6) that "CHL influence was very low in all target seas and time periods. For that reason, the CHL data have been removed from further statistical analyses ...Âż. This contradicts the results of many other studies that have found a successional bloom pattern between diatoms and Ehux blooms, particularly in the Northern Hemisphere (Iglesias-Rodriguez et al., 2002 ; Hopkins et al., 2015 to name a few). I believe this is just an artefact of your analyses which included CHL data up to 4 weeks prior to the Ehux bloom onset, which is too short in many of your target seas where the CHL peak typically occurs 6 to 12 weeks prior to the Ehux bloom. I suspect you will find different results if you extended your time window to 12 weeks prior to the Ehux bloom inset to include the CHL peak.

Finally, the FFs considered in your analyses are now much reduced to just five factors: temperature, salinity, Light availability (PAR), Ekman Layer Depth, and surface current speeds. This should be expressly put in your abstract as a limitation of the approach.

Technical comments

P1, L13: (Abstract) "is known" suggest to be more careful and replace by "is thought".

P2, L14: "tendency to proliferate poleward" was recently clearly demonstrated in the Northern Hemisphere by Neukermans et al. (2018, Glob Change Biol. 2018;24:2545–2553, DOI: 10.1111/gcb.14075)

P2, L18-23: broadly divided into two groups, viz. vector and scalar factors; not clear

P6, L13-17: Rivero-Calle et al. (2015, Science, 10.1126/science.aaa8026) has also used Random Forests in a study to examine the importance of over 20 forcing factors of Ehux in the North Atlantic over the past 50 years. She has made an extensive comparison between the different statistical techniques, and showed that among the multitude of multivariate statistical approaches RF came out as best performing. I believe a reference to her work would be appropriate, and may help you strengthen

your choice for RFs.

P6, L25: Reference to Iglesias-Rodriguez et al. 2002 would be appropriate. Iglesias-Rodrĭ's'guez, M. D., C. W. Brown, S. C. Doney, J. Kleypas, D. Kolber, Z. Kolber, P. K. Hayes, and P. G. Falkowski, Representing key phytoplankton functional groups in ocean carbon cycle models: Coccolithophorids, Global Biogeochem. Cycles, 16(4), 1100, doi:10.1029/2001GB001454, 2002.

P11, L16-20: too speculative

Figures:

Legend to figure 1, please define the categories (i)-(iii) and explain boxes and whiskers extent

Figure 3: Given the huge number of data shown in your scatter plot, a density plot would be more appropriate. Why do you have a bimodal SST – CC relationship in the Labrador Sea?

A map of your study regions would be very useful.

Figure 4: why do you only show pastcast results for half of your study areas?

[Figure]

---

## Referee Comment (RC2) · Anonymous Referee #2 · 22 May 2019

The manuscript by Kondril and colleagues, describes a performant methodology to predict the surface and the intensity of the blooms of Emiliania huxleyi from physical oceanographic parameters (SSS, SST. . . ) that can be estimated from satellite imagery. This is very interesting because it would permit to test which parameter is pertinent for blooms and to test future impact of climate change on coccolithophore blooms in those area. The methodology is based on Random Forest. This is a powerful tool. The results of this methodology seems very promising since it unable the prediction of the bloom quite efficiently. Therefore this work suitable for publication in BG.However I do

not recommend publication in its present form and suggest major revisions, and this for 4 mains reasons :

1- Beside the methodology advances, we do not learn enough on the blooms itself. For exemple the manuscript does not describe what are the conditions that drive a E. huxleyi bloom. I understand the Random Forest is not the tool developed for that. But it could be turn as a diagnostic tool by playing with and/or changing some parameters in the model in order to describe the effect of those changed parameters. For example it could be tested what are the consequence of a temperature change of 1°C on the bloom repartition. At present, the only added knowledge on the bloom is that 'E. huxleyi phytoplankton were highly adaptive to the environmental conditions and capable of arising and developing within a wide range of FFs' (P10 L6). This is quite disappointing because it is known already, and express the absence of the understanding on the reason of those bloom. It is therefore difficult to see what this paper add to our current knowledge of the E.huxleyi blooms.

2- An interesting discussion should be made on the surprising discovery that the bloom can be predicted with only physical parameters and not chemical ones. In particular, often the nutriment concentration or the carbonate chemistry are seen as important factors for predicting blooms (for example in upwelling area). Here the blooms can be predicted without those. This is diagnostic of some peculiar response of the phyto-plankton to the physical condition solely. This discussion is missing.

3- The figures are often are not informative enough: For example what are the infor-mations provided by the large scatters in SST/concentration in Fig. 3. Similarly the overlaps shown in Fig. 2 are non informative. How the 3 maps showing the predicted models quality in Figure 4 have been chosen ?

4- The manuscript is poorly written with many sprawling sentences. The terms used are often vagues with the common use of positive or qualitative expression such as 'highly consequential', 'gigantic'. Often the sentence are not precise : (e.g. 'E. huxleyi is capable of affecting both the marine ecology and carbon fluxes at the atmosphere-ocean interface' ) (without saying in which direction it affects marine ecology and carbon fluxes and in which quantity).

In conclusion this manuscript present a new and powerful methodology without its application. The manuscript should be written in a more concise manner and with more precisions.

Some minor suggestions bellow: -P1L9 replace 'a coccolithophore E. huxleyi' by 'the coccolithophore E. huxleyi'

-P1 L13 remove 'that is known to significantly'

-P1 L14 remove 'and can be retrieved from remote sensing data'

-P1 L22 write each sea target or each targeted sea ?

-P1 L30 THE (not a) unicellular

-P1 L31 remove 'veritably'

-P2 L2 remove 'the waters with the trophic status within the' replace range by waters.

-P2 L6 find another way (with less emphasis) to write this obscure sentence 'This assertion is substantiated by the typical scale of the E. huxleyi phenomenon'

-P2 L7 remove 'gigantic' and add 'large' between 'covering' and 'marine'

-P2 L17 explain 'forward and feedback mechanisms.'

-P2L31 replace 'while in real life' by 'while in natura'

-P2 L32 replace 'strains' by 'morphotypes'

-P3 L20 'the above stated vision' Vision is presumptuous.

-P3 L23-25 remove because not necessary

-P10 L20 'highly consequential' : precise how.

-P11 L10-12: Explain in details this. Because it appears to be very important but not developed enough.

---

## Author Comment (AC1) · 28 May 2019

We appreciate the reviewer's interest in our research, and below there are our reciprocate comments.

1. The first two bullets of the review regard the use of PAR vertical profiles within MLD in light of 8-day MLD data availability on a global scale as it was done by e.g. Lacour et al., 2015. However, Lacour et al. employed to this end monthly climatological data interpolated to an 8 day time period (incidentally, previously we published a description of such databases with a specification of their merits and shortcomings: Kondrik et al., 2017). Nevertheless, the use of the aforementioned data is at odds with the very essence of the methodology of bloom causal factors identification: climatological parameters do not vary from year to year, and thus do not reflect their status on concrete years. Moreover, as the intraannual timing of E. huxleyi blooming varies within 1-2 weeks falling on one and the same month(s), the application of climatological data on PAR vertical distributions would mean that ones and same MLD profiles would be used (e.g. August-September in the case of the Barents Sea) with the only varying parameter -PAR at the surface, but this is the parameter that we actually exploited. This parameter veritably serves the major purpose of our research, namely, the establishment of interannual variations in E. huxleyi blooms for each target sea. A climatological approach would not be appropriate.

It should be also noted that even in a hypothetical case of actual MLD data availability, the utilization of such data for restoration of PAR vertical profiles would be highly problematic if not impossible. Indeed, to restore the vertical profile of PAR within MLD requires in turn data on all optically active components co-occurring with E. huxleyi cells and scales/coccoliths. Understandably, such a requirement is impractical.

2. Regarding the issue of E. huxleyi morphotypes. We agree with the reviewer that such data on morphotypes would be useful for interpretations. However, such information is unavailable for all seas targeted in our study, and we abstained from discussing this option, confining our discussion exclusively to those factors the data on which was confidently established.

3. The issue of using the Takashashi data on pCO2. We are certainly aware of these data as well as those in LDEO and SOCAT data (Kondrik et al., 2018). However, again the Takashashi data are climatological, and hence no interannual variations in pCO2 could be retrieved from them. As to the original/ in situ data on pCO2, on which the SOCAT database is developed, their size reduces drastically upon extraction of the information on our concrete seas and bloom periods. The resultant amount of data

becomes insufficient for use in machine learning technology applied in this study.

4. The reviewer is absolutely right that typically E. huxleyi blooms are preceded by some intense growth of native non-calcifying alga like diatoms (most often cases). The moments between the outbursts (i.e. maximum growth) of such cold-water photosynthesising alga and E. huxleyi vary but usually are about three to four weeks (sometimes a bit longer). In some cases the onset of E. huxleyi blooms occur while the "tail" of preceding bloom (for simplicity reasons, let's call it "diatomic bloom") overlaps the beginning of the E. huxleyi bloom period (e.g. Pozdnyakov, et al., 2017); in such circumstances the concentration of diatoms might constitute about 10% of the concentration of E. huxleyi) [e.g. Lavender et al., 2008]. Our observations reported in the submitted paper have actually revealed (through the concentration of chlorophyll as a proxy) the presence of "diatomic" alga within a two or four week period prior to the maxim of E. huxleyi growth. However, no indications were revealed that the preceding "diatomic" bloom somehow affected the extent/intensity of the following bloom of E. huxleyi. So that, let us repeat for clarity: it is not the question that our observations failed to detect the presence of "diatomic" phytoplankton, but that there was no evidence that the chemical "preparation" of surface water (in terms of N:P ratio) by the preceding bloom was appreciably consequential for the development of E. huxleyi bloom.

All other critical remarks contained in Section "Specific comments" will be accepted in the final revised version of our paper.

References

Lacour, L., H. Claustre, L. Prieur, and F. D'Ortenzio (2015), Phytoplankton biomass cycles in the North Atlantic subpolar gyre: A similar mechanism for two different blooms in the Labrador Sea, Geophys. Res. Lett., 42, doi:10.1002/ 2015GL064540.

Dmitry Kondrik, Dmitry Pozdnyakov & Lasse Pettersson (2017) Particulate inorganic carbon production within E. huxleyi blooms in subpolar and polar seas: a satellite time series study (1998–2013), International Journal of Remote Sensing, 38:22, 6179-6205,

DOI: 10.1080/01431161.2017.1350304

Takahashi, Taro; Sutherland, Stewart C; Wanninkhof, Rik; Sweeney, Colm; Feely, Richard A; Chipman, D W; Hales, Burke; Friederich, G; Chavez, Francisco P; Watson, Andrew J; Bakker, Dorothee C E; Schuster, Ute; Metzl, Nicolas; Yoshikawa-Inoue, Hisayuki; Ishii, Masao; Midorikawa, Takashi; Nojiri, Yukihiro; Körtzinger, Arne; Steinhoff, Tobias; Hoppema, Mario; Ólafsson, Jón; Arnarson, T S; Tilbrook, Bronte; Johannessen, Truls; Olsen, Are; Bellerby, Richard G J; Wong, Charles S; Delille, Bruno; Bates, Nicolas R; de Baar, Hein J W (2009): Climatological mean and decadal change in surface ocean pCO2, and net sea–air CO2 flux over the global oceans. Deep Sea Research Part II: Topical Studies in Oceanography, 56(8-10), 554-577, https://doi.org/10.1016/j.dsr2.2008.12.009

Kondrik, D. V., Pozdnyakov, D.V., Johannessen, O. M. 2018. Satellite evidence that E. Huxleyi phytoplankton blooms weaken marine carbon sinks. Geophysical Research Letters. doi: 10.1002/2017GL076240.

Pozdnyakov D.V., Pettersson L.H., Korosov A.A. (2017). Exploring the Marine Ecology From Space. Springer International Publishing Switzerland. doi: 10.1007/978-3-319-30075-7.

Lavender S.J., Raitsos D.E., Pradhan Y. (2008) Variations in the Phytoplankton of the North-Eastern Atlantic Ocean: From the Irish Sea to the Bay of Biscay. In: Barale V., Gade M. (eds) Remote Sensing of the European Seas. Springer, Dordrecht

---

## Author Comment (AC2) · 28 May 2019

We thank the referee #2 for his/her comments, and below are our answers.

1. First of all, the present work is intended to investigate the influence of remotely retrievable co-acting factors conditioning the growth of E. huxleyi in real-life conditions. I.e. it is not intended to investigate the individual influence of changes in acting alone parameters on E. huxleyi bloom dynamics (such studies were performed in laboratory conditions by many authors with regard to different parameters, and we provided the

[Figure]

relevant references in the manuscript). To achieve the goal posed above, it was impossible to employ in situ data. Indeed, shipborne data even from an unrealistic large scale field campaign would not provide the desired data at a required high spatial and temporal resolution within a multidecadal time period. Understandably, investigations in laboratory /mesocosm conditions are unable to prove the sought for data on a collective influence of a significant number of co-acting causal factors, leaving alone the fact that data from laboratory/mesocosm conditions could not be regarded as faithfully reflecting the real life conditions. And of course they would not allow to investigate the interannual dimension of this phenomenon. That is why we resorted to satellite data: this approach meets the goal of reflecting the actual/real-life co-action of causal factors on E. huxleyi. blooms. Importantly, the large time period and a variety of target seas make the results obtained unapparelled. Our results are in no way limited solely by ascertaining the sea-specific causal factors prioritization in terms of their influence on E. huxleyi blooms. No similar results have ever been published before. We came up with a wealth of other findings/establishments (such as e.g. the concatenated ranges of causal factors values corresponding to pick blooms in the target seas as established over nearly 20 years, etc.) that are summarized in Conclusions. We don't think we should reiterate them here.

2. In addition to what is given in Conclusions, we draw the reviewer's attention that together with the established prioritizations the development of reliable RFC models opens the way to predict the future tendencies in E. huxleyi dynamics in conditions of ongoing climate change. Indeed, employment of both CMIP5 models predictions of the sets of prioritized causal vector and scalar factors changes over the 30 forthcoming years within the target seas and the RFC models for each respective sea, it is allows to assess the expected tendencies in the future developments of E. huxleyi-driven phenomena. This work is presently being done by us. So, at this further stage of research, it is possible to accomplish what the reviewer meant in his/her remark.

3. As for the figures, the respective interpretation and reference are given in the

manuscript according to the Biogeosciences policy: the main message of figure 3 is given in P10 L21-24, figure 2 - P8 L3-9. Please note that its purpose is not at all to illustrate "overlapping" but to display the difference in the statistical ranges of FFs inherent in the cases of large and small blooms. The three maps were chosen to show the most extensive blooms occurred in the studied seas to better compare the modelling results.

4. The reviewer's stylistic remarks will be closely considered in preparing the revised manuscript.

We thank the reviewer for the suggestion regarding the possible application of the study performed. In Conclusion we twill point out that the results obtained will be used (as we explained in bullet 2) for the projections of E. huxley bloom temporal and spatial tendencies up to 2050 by using the CMIP5 vector and scalar modelling results in combination with our RFC models.

---

## Author Comment (AC3) · 10 Jun 2019

Associate Editor Decision: Reconsider after major revisions (28 May 2019) by Jean-Pierre Gattuso Comments to the Author: Dear Author,

The referees rated your manuscript below the level of quality expected in Biogeosciences papers. Both reviewers recommend to reconsider the manuscript after major revisions. I urge you to address their comments thoroughly. For example, the issue of the possible contribution of the carbonate chemistry in driving blooms and distributional

changes of Emiliania huxlehi cannot be summarily dismissed as is done in your reply to the comments. There are ways to estimate changes in the carbonate chemistry when observational data sets are poor, for example Bittig et al. (Frontiers in Marine Science) and Denvil-Sommer et al. (Geoscientific Model Development).

The revised manuscript will undergo a second round of review to ascertain that all comments and concerns have been satisfactorily addressed.

Sincerely, Jean-Pierre Gattuso BG editor

————————————————

Dear Prof. Gattuso, First of all, thank you for your close consideration of our manuscript applied to the Biogeosciences journal (Kondrik et al., Prioritization of the vector factors controlling Emiliania huxleyi blooms in subarctic and arctic seas: A multidimensional statistical approach, https://doi.org/10.5194/bg-2019-104). In this letter we will try to clarify the question emerged from the public discussion of our manuscript related to the use of data on carbonate chemistry. This question was initially raised by Dr. Neukermans (referee #1). Indeed, our studied variables did not include the parameters, directly connected to the carbonate chemistry system, as this work was done with the use of spaceborne data. Of course, addition of such parameters, as background pCO2, would be preferable. Datasets/approaches containing this parameter were proposed by Dr. Neukermans (Takahashi pCO2 climatology) and you (Bittig et al., 2018; Denvil-Sommer et al., 2018). The first dataset was discussed by us in the reply to the Dr. Neukermans, so here we will focus on the latter two approaches. The proposed approaches are indeed very interesting in terms of both methodology and results, but meet some difficulties inherent in our specific study. In particular, work performed by Bittig et al. (2018) has some limitations described by the authors themselves in (Bittig et al., 2018), which appear to be crucial when implementing its results to our RFC algorithm: 1) The results have an increased bias at the surface layer (p. 15). 2) The decoupling is noticeable following intense blooms or long bloom periods (p. 15). It is

also worthwhile to mention that this approach has not been specifically tested in the coccolithophore bloom regions, which are known to alter the carbonate chemistry very significantly (this issue will be discussed below). 3) It is clearly stated that CANYON-B and CONTENT algorithms have a clear focus on the water column and ocean interior variable estimation (p. 15), which, in turn, can lead to the uncertainties in some results (they can be seen in, e.g. Fig. 8 of (Bittig et al., 2018), constituting up to 50 $\mu$atm in comparison to the Polarstern vessel data). 4) In caption to Figure 9 of (Bittig et al., 2018) it is also stated that CONTENT pCO2 estimations can be lower than actual SOCAT data (and even climatology) in the high latitude North Pacific during Summer, which supports our statement that in our study regions (polar and subpolar latitudes of Atlantic, Pacific and Arctic ocean) practically all climatologies/algorithms can give much higher errors due to low amount of testing in situ data. Other work proposed by you is also very interesting, but also has its limitations specifically for our study. This approach is very similar to reanalysis, as it employs the climatological data as the first step and then implements the neural networks to assimilate the SOCAT pCO2 data. It leads us to the same conclusion: whereas this approach has, no doubt, a lot of applications on global scale, in subpolar and polar regions it practically shifts to the same climatological dataset, with only slight local changes. Thus, the discussed datasets/approaches have their limitations/uncertainties, which due to their nature of data availability tend to increase poleward, which, in turn, lead us to question their adequacy for our study. But, again, these limitations do not lower the significance of discussed works, but only underpin the difficulties of studying the polar and subpolar regions in terms of available and reliable data. But even more important obstacle for application of discussed approaches aimed at estimation of changes in the carbonate system lies in the fact that in our case E. huxleyi, as calcifying alga can, drastically change the pCO2 in water - up to hundreds of $\mu$atm, which relates to 60% of background pCO2, according to our estimates in (Kondrik et al., 2018). In addition, Shutler et al. (2013) report on an average reduction in the monthly air-sea CO2 flux by about 55% across the marine tracts encompassing extensive E. huxleyi blooms in the North Atlantic, whereas the

maximum reduction over the time period 1998–2007 was registered at 155%. Due to the fact that RFC is developed for estimation of weights/importances of forcing factors without the determination of (in terms of, e.g. pCO2) sources of changes in these parameters (background or caused by the calcification process), we can't interpret the resulting importance of this parameter in terms of carbonate system's status quo during the bloom period. In simple words, the RFC just can't tell us whether it is pCO2 changes are influencing on the E. huxleyi blooms or vice versa. Moreover, the "artificially" increased importance of pCO2 (we are confident that it will be high taking into account above stated facts) can significantly decrease the importance of all other parameters, as the resultant importances are relative and always have 100% in total, which, in turn, will lead us to much worse modelling results. This means that pCO2 (as well as the other variables related to the carbonate chemistry described in (Bittig et al., 2018)) can't be employed for the RFC training because the E. huxleyi blooms phenomena itself has a very strong influence (if not prevailing) on the carbonate system state inside the E. huxleyi bloom areas. At the same time, being initially consistent (as we showed through a 2-decadal time series) our spaceborne variables can arguably give access to the carbonate system characterization through the water chemistry theories, as it equally refers to the four variables discussed by Bittig et al. (2018). We thank you again for all your efforts made up to this point for improving our manuscript (as well as the two referee's) and hope that all above stated arguments can help to solve this whole situation.

On behalf of the co-authors, Prof. Dmitry Pozdnyakov

———————————————————

Dear Colleague, I quickly reply to your message. The decision to request a major revision is based on the reviewers' evaluation and my own and remains unchanged. I have forwarded to you the scores, which are below the level of quality expected in Biogeosciences. As my decision letter mentions, the issue of the carbonate system is not the only argument. Other criticisms and suggestions from the reviewers must be satisfactorily addressed, that is in a better way than in your replies to the comments. The style also needs a lot of attention. As you acknowledge, considering pCO2 (I would say the status of the carbonate system) is highly relevant. It is known to be a driver for coccolithophores, even though species and strains do seem to exhibit different sensitivities. The data coverage may be inadequate in space in time to be used in your statistical and modelling approaches but that is not a good reason to dismiss these variables. It makes your conclusion that "the adequacy of the developed models for FFs prioritization with regard to E. huxleyi blooms" very weak and questionable. Your study aims at identifying the variables which control the extent and magnitude of E. huxleyi blooms. The carbonate chemistry may play a big role in the onset of the bloom. Your argument (II) that Emiliania raises pCO2 does not apply there. I hope this helps. Kind regards, Jean-Pierre Gattuso

————————————————————

Dear Colleague,

We send our manuscript with the additions and changers required by reviewers. As far as in your letter of June 1 you practically ignored/waved away our reasonings without any real consideration, we rely on it that the revised text will be given full consideration by the two reviewers who have already commented on our study and were given our responses/clarifications. We earnestly hope that their criticism, if there is any, will be seriously argued but not reduced to a mere dismissal as it was in your letter. Together with the revised manuscript we also submit our previous letter to you using the option "Authors' comments" just to let the two reviewers get aware of our argumentation given to you regarding the inappropriateness of using in our analyses the datasets by Bittig et al, 2018) and Denvil-Sommer et al. 2018). Kind regards, Co-authors